# Dengue transmission dynamics prediction by combining metapopulation networks and Kalman filter algorithm

**Qinghui Zeng[1], Xiaolin Yu[1], Haobo Ni[1], Lina Xiao[1], Ting Xu[1], Haisheng Wu[1], Yuliang Chen[2], Hui Deng[3], Yingtao Zhang[4], Sen Pei[5]\*, Jianpeng Xiao[6]\*, Pi Guo[1,7]\***

**1** Department of Preventive Medicine, Shantou University Medical College, Shantou, China, **2** Department of Medical Quality Management, Nanfang Hospital, Guangzhou, China, **3** Institute of Vector Control, Guangdong Provincial Center for Disease Control and Prevention, Guangzhou, China, **4** Institute of Infectious Disease Control and Prevention, Guangdong Provincial Center for Disease Control and Prevention, Guangzhou, China, **5** Department of Environmental Health Sciences, Mailman School of Public Health, Columbia University, New York, New York, United States of America, **6** Guangdong Provincial Institute of Public Health, Guangdong Provincial Center for Disease Control and Prevention, Guangzhou, China, **7** Guangdong Provincial Key Laboratory of Infectious Diseases and Molecular Immunopathology, Shantou, China

\* sp3449@cumc.columbia.edu (SP); xiaojp@gdiph.org.cn (JX); pguo@stu.edu.cn (PG)

**Data Availability Statement:** To protect the privacy of study participants, primary data are not publicly available. To access these data and/or to seek permission for its use, please contact the

## Abstract

Predicting the specific magnitude and the temporal peak of the epidemic of individual local outbreaks is critical for infectious disease control. Previous studies have indicated that significant differences in spatial transmission and epidemic magnitude of dengue were influenced by multiple factors, such as mosquito population density, climatic conditions, and population movement patterns. However, there is a lack of studies that combine the above factors to explain their complex nonlinear relationships in dengue transmission and generate accurate predictions. Therefore, to study the complex spatial diffusion of dengue, this research combined the above factors and developed a network model for spatiotemporal transmission prediction of dengue fever using metapopulation networks based on human mobility. For improving the prediction accuracy of the epidemic model, the ensemble adjusted Kalman filter (EAKF), a data assimilation algorithm, was used to iteratively assimilate the observed case data and adjust the model and parameters. Our study demonstrated that the metapopulation network-EAKF system provided accurate predictions for city-level dengue transmission trajectories in retrospective forecasts of 12 cities in Guangdong province, China. Specifically, the system accurately predicts local dengue outbreak magnitude and the temporal peak of the epidemic up to 10 wk in advance. In addition, the system predicted the peak time, peak intensity, and total number of dengue cases more accurately than isolated city-specific forecasts. The general metapopulation assimilation framework presented in our study provides a methodological foundation for establishing an accurate system with finer temporal and spatial resolution for retrospectively forecasting the magnitude and temporal peak of dengue fever outbreaks. These forecasts based on the proposed method can be interoperated to better support intervention decisions and inform the public of potential risks of disease transmission.

Data-center of China Public Health Science (https://www.phsciencedata.cn/Share/index.jsp) or email data@chinacdc.cn.

**Funding:** The authors received no specific funding for this work.

**Competing interests:** The authors have declared that no competing interests exist.

## Author summary

Dengue fever is a vector-borne disease transmitted by the Aedes aegypti mosquito and has become a global threat. The increased possibility of individual exposure to disease vectors due to human travel allowed for the rapid spread of dengue virus to new susceptible populations, ultimately resulting in severe febrile illness and substantial disease burden. Therefore, the integration of information on human movement into infectious disease prediction model will contribute to the prevention and control of infectious diseases. This study developed and retrospectively validated a forecasting system for predicting the spatial spread of dengue fever in Guangdong province, China, using population mobility and dengue fever incidence data. Compared with the local forecasting model designed for each individual city, the proposed system generated improved prediction performance with respect to the targets including peak time, peak intensity, and total incidence for retrospective forecasts of dengue fever epidemics in 12 cities of Guangdong province. The proposed method can be potentially adapted to predict other mosquito-borne infectious diseases.

## Introduction

Dengue fever, a mosquito-borne infectious disease, is caused by four viral serotypes and mainly transmitted by *Aedes albopictus* and *Aedes aegypti* [1]. More than 390 million people worldwide are infected with the dengue virus each year, resulting in an estimated global cost of $8.9 billion (95% uncertainty interval 3.7–19.7) [2,3]. Infections of the disease in the Asia-Pacific region bear about three-quarters of the global dengue fever burden [4]. Predictions of the magnitudes and temporal peak of dengue epidemics in advance are crucial for planning and resource allocation for public health initiatives [5]. However, emerging studies on analyzing historical dengue outbreaks reveal a complex problem involving climate impacts, population changes caused by migration patterns, mechanistically complicated transmission rates, time-varying mosquito populations, and underreporting infections [6–14]. Although the factors mentioned above have been separately documented, it still remains unclear how to incorporate all these factors to improve the prediction performance of the magnitude and temporal peak of dengue epidemics.

In China, the outbreak of dengue fever varies greatly across cities and seasons. Specially, few cases were reported in low-disease periods. This means that the incidence of dengue is zero-inflated from a time-series perspective (S1 and S2 Figs). Previous studies have proposed that the use of the ensemble adjusted Kalman filter (EAKF) assimilation algorithm in conjunction with a compartmental model could solely capture the trend of local infectious disease transmission [15–20]. However, these models have not been able to track epidemics and outbreaks of dengue in terms of spatial transmission. Models only consider the time trend of dengue in a single geographic location. As previous research has pointed out, spatial diffusion is vital for understating the evolution of real-world complex dynamical transmission of mosquito-borne diseases, and human mobility plays an important role in the spatial spread of diseases [14,21]. Therefore, in order to more accurately track the spatial spread trend of dengue, a promising solution is to propose a novel meta-population network based on the previous SIR-EAKF model, which can not only consider the temporal epidemics of dengue but also track the spatial spread trend.

Additionally, limited by passive disease surveillance, difficulties in recording asymptomatic or mild dengue infections may lead to underreporting of infections [22]. Network dynamic models based on human mobility data can partially alleviate the bias [23]. For this reason, the

study aims to construct metapopulation networks incorporating multiple factors (climate impacts, population changes caused by large spatial scale real-time human migration patterns, mechanistically complicated transmission rates, time-varying mosquito populations, and underreporting infections) for the forecasts of the magnitude and temporal peak of dengue epidemics, in which the filtering techniques iteratively update, or adjust model simulation estimates of the dynamic state.

Therefore, a combined forecast system based on metapopulation networks and the Kalman filter algorithm was developed to generate weekly real-time forecasts of dengue. As shown in Fig 1, the apparent spatial movement of dengue during the 2019–2020 epidemics was observed. Our results demonstrated that the posterior fitting captured the weekly dengue incidence and retrospectively validated the prediction accuracy of the model regarding to three targets including peak time, peak intensity, and total incidence for 21 cities in Guangdong

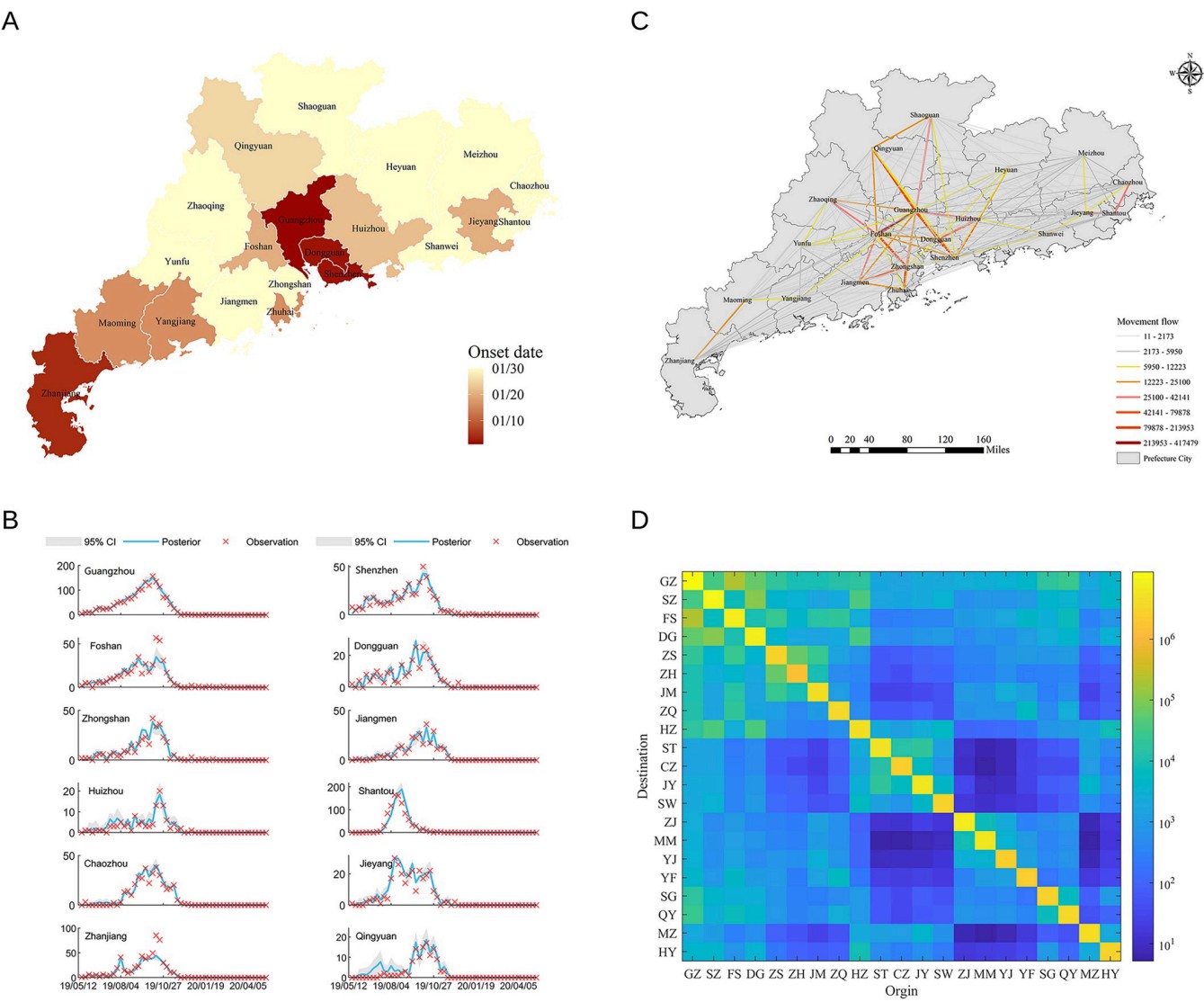

**Fig 1. The spatial transmission of dengue fever in Guangdong province, China, in the 2019 pandemic season.** (A) Onset date of 21 prefecture-level cities in Guangdong province. (B) Mobile user flow between cities derived from mobile phone data. (C) The number of new cases of dengue fever per week (cross symbol) in each city. The solid line and shadow area are the posterior mean and 95% confidence interval (CI) of the metapopulation networks–EAKF fitting, respectively. (D) The daily average population movement between 21 prefecture-level cities in Guangdong province is based on mobile phone trajectory records. Base map sourced from the DIVA-GIS (https://www.diva-gis.org/gdata).

province, China. The method exhibits an impressive performance in predicting dengue spread dynamics. In addition, the system was utilized to assimilate model-simulated outbreaks to obtain time-varying inferred values of the key parameters, which further verified the capability of the model's parameter inference. This reliable dengue prediction approach can help design effective public health interventions to control and prevent dengue fever.

## Materials and methods

### Data sources

The current study involved assimilation and analysis of 21 prefecture-level cities to build a spatial-temporal prediction system for retrospective epidemic prediction. These cities in Guangdong province of southeastern China have long been plagued by frequent outbreaks of dengue fever [24].

The surveillance data of dengue cases from 2015 to 2020 were obtained from Guangdong Center for Disease Control and Prevention (Guangdong CDC). All cases were diagnosed according to the dengue fever diagnostic criteria issued by the Chinese Ministry of Health (WS216-2008) [25]. In particular, all the four dengue virus serotypes are included in the statistics and summarized by week. However, since there were too few observations of dengue fever during the 2015–2017 dengue epidemic seasons, we limited the retrospective prediction range to two consecutive dengue seasons from 2018 to 2020. Afterward, there are only sporadic cases in some cities (such as only 1 case in 2019 in Shanwei city and 6 cases in 2019 in Shaoguan city). From the perspective of time series analysis, these sporadic cases can basically be treated as random numbers. The number of cases is very sparse and it is difficult to model the disease transmission using existing methods. Therefore, 12 prefecture-level cities in 2019 were selected for retrospective prediction to ensure that the assimilation algorithm has enough dengue surveillance data for model calibration. In addition, according to the strategy in precious study to reduce the effect of filtering divergence [26], we also precisely defined the dengue season as the 19th week of the former year to the 20th week of the succeeding year.

Mosquito oviposition index (MOI) is currently widely used by Guangdong CDC for mosquito density monitoring. It is the percentage of *Ae. albopictus* positive ovitraps in all ovitraps collected from a specified area, and reflects the abundance of the adults [27,28]. Initially, the prediction model needs to obtain the time-varying time series of the mosquito population, so we used MOI data to quantify the natural birth rate of the mosquito population. We used a revised version of the birth rate calculation formula established by Chen [26]. Here, missing values were filled by data from surrounding cities, and subsequently a complete mosquito vector trend was provided. Because MOI data do not differ significantly among geographically bordering cities due to similar climatic conditions in Guangdong province, subtle fluctuations in time series might still be ignored by the data missing value filling method.

Daily ambient temperature for the same period was publicly available on the China Meteorological Data Sharing System (http://data.cma.cn/). The association between dengue virus reproduction in mosquito vector saliva and ambient temperature has been reported in previous studies [29], and this association has been expressed by a formula [30]. The function is listed as follows:

$$f(T) = \begin{cases} 0 & T \leq 18 \\ (T-18) \times 8/3 & 18 < T \leq 23 \\ 40/3 + (T-23) \times 16/3 & 23 < T \leq 28 \\ 40 + (T-28) \times 20/3 & 28 < T \leq 32 \\ 200/3 & T > 32 \end{cases} \quad (1)$$

where $T$ is the daily ambient temperature, and $f(T)$ is the population transmission rate. This study used the same formula to estimate the probability of dengue virus transmission from mosquito bites based on meteorological data. Although observations from multiple monitoring stations in the same city may not be precisely identical, the differences are immaterial because the variables used in the study were obtained by calculating the average temperatures for each week in each city.

To assess intercity population migration, we analyzed all citizens' mobile phone data which have been proven in previous studies to well estimate population mobility patterns [8,30]. In this study, we used data from the China Unicom, a company that operates domestic and international communications facilities, to study mobility patterns from 1 January to 31 December 2018. The China Unicom company holds about 19% of China's market share [31] and is one of the three largest telecom operators in China, which has extensive geographic coverage. All data collection and processing were performed on the operator's servers to protect users' privacy during data pre-processing. Subsequently, the users' cities were returned based on the cellular tower locations. We later performed further spatial and temporal data aggregation based on prescribed city geocodes. For each perfecture-level city, we represent the movement of individuals between cities as a bipartite network with time-varying edges, in which the weight of an edge between cities represents the number of visitors from that city to the next during a given day (S1 Video).

Because the retrospective predictions in this study included two consecutive dengue seasons from 2018 to 2020, it is assumed that both dengue seasons have the same pattern of population movement. In addition, the total resident population of each city was obtained separately from the statistical yearbook. Based on the above assumptions and data, the matrix of 2018–2020 daily human mobility was derived for subsequent model operations.

## Metapopulation network

To retrospectively predict the spatial dissemination of dengue virus, we built metapopulation networks shown in Fig 2 that used disease monitoring data from multiple cities to flexibly generate spatial transmission patterns for reliable ensemble forecasts. The metapopulation model connects all cities using human movement. Therefore, the total population can be divided into several subpopulations, each of which represents human migration from the former region to the latter [32].

We overlay a standard mosquito-borne susceptible-infected-recovered (SIR) model modulated by local ambient temperature conditions [33] (S1 Text, temperature and mosquito vector-driven SIR model for dengue fever) on each human mobility network, in which each subgroup maintains its own susceptible (S), infectious (I) and removed (R) states. New infections occurred at all subpopulations, with the metapopulation movement network governing how subpopulations from different cities interact as they visit another one. In our previous study [26], we developed a dengue-specific model based on the characteristics of mosquito-borne transmission. The model consisted of different compartments corresponding to the disease stages of mosquitoes and humans in the subpopulation. According to the transmission characteristics of each compartment, we constructed differential equations, which incorporated the annual average time series of mosquito birth rate ($\mu_b(t)$) and population transmission rate ($\tau_n(t)$). The time-varying $\mu_b(t)$ and a fixed death rate ($\mu_d$) can simulate the natural growth of mosquito population. The model parameter $\tau_n(t)$, estimated by temperature, can explain the probability of successful transmission of dengue virus after a mosquito bite. Afterward, the equations were integrated to obtain the dengue virus transmission between mosquitoes and humans in this thoroughly mixed population, yielding the number of susceptible, infected, and recovered individuals in mosquito population and human population at each time point, respectively (S1 Text, temperature and mosquito vector-driven SIR model for dengue fever).

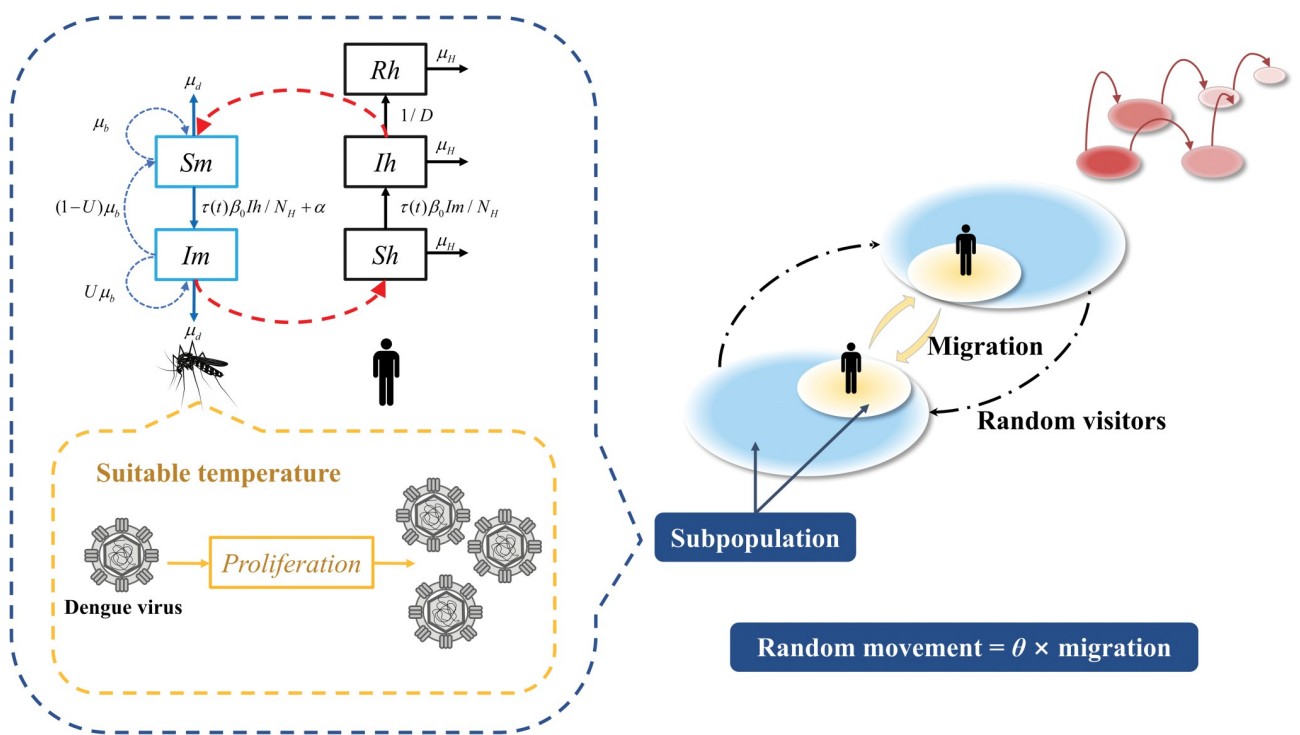

**Fig 2. Model structure.** Blue and black rectangles represent mosquito and human subpopulations, respectively. The yellow compartments represent the proliferation of dengue virus in mosquitoes at the suitable temperature. The right chart shows the human movements structure of the crowd among the various subpopulation.

Consequently, we can fit compartmental models into each edge to better project how the dengue spreads within the subgroups.

From the structural point of view, we simulate irregular movement besides population mobile data. These individuals follow the Markov process and circulate among subpopulations, resulting in population exchange among all subpopulations. Our model assumed that the number of random visitors between cities is proportional to the average population movement between cities. More population mobility meant that there should be more solid links or other incentives to promote the communication of random travelers. To estimate random movement population, we assumed an adjustable parameter $\theta$ (random movement ratio), thus, the random visitors between two sites were given by the product of the average population movement and $\theta$. The $\theta$ is regarded as a threshold at which the random movement is still progressing. Besides, all travelers flow directionally between cities every day, so the volume of travelers in opposite directions between two cities might be different in the origin-destination matrix in this study. When they flow into the destination, they will be allocated to different compartments according to their disease course. This population exchange exists between all paired cities. During the course of disease transmission, each individual may progress across the compartments, with the rate illustrated by these ordinary differential equations:

$$\frac{dS_{nH}^k}{dt} = -\frac{\tau_n(t)\beta_0 S_{nH}^k I_{nM}}{N_{nH}} + \theta\frac{N_{nH}^k}{N_{nH}}\sum_{j\neq n}\bar{N}_n^j\frac{S_{jH}}{N_{jH}} - \theta\frac{S_{nH}^k}{N_{nH}}\sum_{j\neq n}\bar{N}_{jH}^n$$

$$\frac{dI_{nH}^k}{dt} = \frac{\tau_n(t)\beta_0 S_{nH}^k I_{nM}}{N_{nH}} - \frac{I_{nH}^k}{D} + \theta\frac{N_{nH}^k}{N_{nH}}\sum_{j\neq n}\bar{N}_{nH}^j\frac{I_{jH}}{N_{jH}} - \theta\frac{I_{nH}^k}{N_{nH}}\sum_{j\neq n}\bar{N}_{jH}^n$$

(2)

where $S_{nH}^k$, $I_{nH}^k$, and $N_{nH}^k$ were the number of susceptible, infectious, and the total number of individuals in population movement from city $k$ to $n$, where the subscripts H and M stood for the human (host) and mosquito, respectively. $N_{nH}$ represented the human population in city $n$. $\tau_n(t)$ is the population transmission rate at time $t$, $\beta_0$ is the basic contact rate between humans and mosquitoes, $\tau_n(t)\beta_0$ is the contact rate between humans and mosquitoes at time $t$. $D$ represented the infectious period, which defined how long the infectious individuals recovered from the disease. Since mosquito populations cannot move over long distances on a large scale, it was assumed that mosquito movement only occurred within each city, and there was no cross-city movement of mosquitoes. Here, we demonstrated a model developed to illustrate dengue virus transmission among mosquitoes:

$$\frac{dS_{nM}}{dt} = -(\alpha + \frac{\tau_n(t)\beta_0 I_{nH}}{N_{nH}})S_{nM} + \mu_b(t)(S_{nM} + (1 - U)I_{nM}) - \mu_d S_{nM}$$

$$\frac{dI_{nM}}{dt} = (\alpha + \frac{\tau_n(t)\beta_0 I_{nH}}{N_{nH}})S_M + \mu_b(t)U I_{nM} - \mu_d I_{nM}$$

(3)

Here $\alpha$ was the random rate of dengue fever seeding into the local areas, $U$ was the vertical infection rate constant over an outbreak. $\mu_b(t)$ is the mosquito birth rate at time $t$, $\mu_d$ is the mosquito death rate and is constant over an outbreak. These ordinary differential equations were integrated to calculate the number of new infections in each subpopulation, and the weekly number of new infections in city $k$ was obtained by summing up the total number of subpopulations entering city $k$ during that week.

## Parameter inference

The metapopulation model can generate many spatial diffusion scenarios by changing the parameter $\theta$ which was the ratio of human mobility data to random visitors. Besides, by giving vast amounts of parameters $D$ and $\beta_0$, the model can capture enough outbreak scenarios in each subpopulation. So that the model can be used in conjunction with EAKF to infer key parameters of the model and generate ensemble predictions due to its flexibility. EAKF is a data assimilation technology developed by Anderson [34]. The advantages of the EAKF algorithm in assimilating confirmed case counts and updating parameters according to Bayes' rule have been reported in previous studies [20,35]. Therefore, this data assimilation algorithm was also used in this study to assimilate the weekly incidence cases from CDC for continuous calibration of the model and parameters. In model training, the state variables and parameters in the dynamic model are calibrated repeatedly by existing dengue cases. Then the optimized model is integrated into the future to generate prediction. In practice, the popular trajectory set generated randomly is simulated to generate the probability distribution of prediction results.

The metapopulation system has a high dimensional state vector. In each subpopulation $N_n^k$, six state variables: $S_{nH}^k$, $I_{nH}^k$ and the number of new infections per week, $S_{nM}$, $I_{nM}$, and newly infected mosquitoes were recorded simultaneously and parameters $D$ (duration of infection), $\beta_0$ (basic contact rate between humans and mosquitoes), and $\theta$ (random movement ratio) were shared for all subpopulations. For each data assimilation, the parameters and state variables recorded at the previous moment are calibrated by EAKF, a filtering algorithm for high dimensional systems using finite number of ensemble members. Then, they were used in the dynamics model integration for the epidemic prediction in subsequent moment. The above process was repeated until the end of a retrospective prediction period. After the assimilation, the variables and parameters in the dynamic model of infectious diseases have been calibrated iteratively. Finally, the state variables in each subpopulation were aggregated by city to obtain

week-by-week summary data for each state variable with 300 ensemble members (S1 Text, data assimilation methods). Based on these summary data, point estimates were shown as posterior means and statistical uncertainties were shown as 95% prediction intervals.

To verify the accuracy of parameter inference, we examined whether key model parameters can be accurately recovered from synthetic outbreaks generated by models with known true values. Two synthetic dengue fever seasons with the same parameters were simulated by the metapopulation networks (except $\theta$). To simulate a dengue fever outbreak close to the real situation, the simulated synthetic outbreaks were then generated by adding random observation errors (error variance (weekly infections$^2$+100)/25) to the truth. Then, these synthetic error-laden observations were used to assimilate and estimate parameters in the combined system. Simultaneously, the parameter inference error is obtained by comparing the inferred parameter with the true value (S3 Fig). In addition, to further verify the stability of the model parameter inference performance, we also extended the number of simulations to ten times, and used the metapopulation model to perform parameter inference for these ten simulated outbreaks.

## Retrospective forecasts

We generated 300 independent ensemble forecasts for each week during the dengue season from 2018 to 2020 using both the metapopulation model and the isolated SIR model (S1 Text, temperature and mosquito vector-driven SIR model for dengue fever). Turning to the initialization for crucial state vector, the initial conditions were drawn using a Latin hypercube sampling with the following ranges: $S_{nH}^k \in [0.7N_{nH}^k, 0.9N_{nH}^k]$, $I_{nH}^k \in [10^{-6}, 10^{-4}]$, $S_{nM} \in [N_{nM}]$, $I_{nM} \in [10^{-6}, 10^{-4}]$, $D \in [5,7]$, $\beta_0 = [0.01, 0.1]$, $\theta \in [0, 0.5]$, $N_{nH} = N_{nM}$, $N_{nH}^k$ represented human migration form location k to location n derived from mobile phone data. The observed variable in each subpopulation was initiated as 0 because it could be fully determined by other state variables and parameters during model integration. All subpopulations share the same parameters $D$ and $\theta$. Besides, some state vectors were initialized with constant parameter: $U = 0.25$, $\alpha = 1/500000$ and $\mu_d = 1/15$ over the whole dengue seasons. For each subpopulation, the susceptible population, the infected population and observed incidence for both mosquitos and human were recorded in the state vector. We subsequently obtained prediction accuracy of predicted targets (peak week, peak intensity, and total incidence) relative to true values for 12 cities in Guangdong province. The remaining nine cities were excluded, as they were not well represented in the CDC dengue dataset due to the low number of dengue records (S1 and S2 Figs) [36]. In addition, since dengue outbreaks often begin in June and end in December of each year, a dengue season was defined as week 19 of each year to week 20 of the immediately following year in order to reduce the impact of filtering divergence caused by assimilating many zero-inflated data in the earlier stages of assimilation. (S4 Fig).

## Results

As previously described, this study aimed to construct metapopulation networks using human mobility data to predict the spatial and temporal transmission trends of dengue viruses, and produce weekly real-time predictions of the magnitude and temporal peak of the epidemic. Initially, we selected dengue fever observations from 12 prefecture-level cities in Guangdong province in 2019 for assimilation and we publicly obtained the raw shape files from the DIVA-GIS (https://www.diva-gis.org/gdata), and generated the maps in this study. Among these dengue cases, we observed a significant spatial movement of dengue fever in 2019 (Fig 1A), with an earlier dengue epidemic in the Pearl River Delta and a temporal lag in other regions. In particular, in order to construct the metapopulation networks, we used the matrix of daily population migration data. We found that population movements were more

numerous and frequent in the Pearl River Delta region based on mobility data (Fig 1B) and that most population movement events occurred between neighboring cities, while long-distance movements were quite rare (Figs 1D and S5). To verify the relationship between dengue epidemic and population movement, we used the empirical dynamic modeling (EDM) method [37] to reconstruct the dengue cases with 3 time series (mosquito vector MOI, temperature, and population movement). A detailed analysis of the relationship between these 3 time series and the occurrence of dengue fever had been performed, and the results showed that the 3 time series' cross mapping skills were positive except for few cities (S6 Fig), suggesting that the time series of mosquito vector MOI, temperature, and population migration are positively correlated with incidence time series of dengue fever. Therefore, we employed the forecast system based on the metapopulation SIR model and EAKF algorithm to fit actual observed data of dengue cases in Guangdong in 2019 (S7 Fig for the 2018 forecast results). The forecast system produced posterior fitting of the epidemic shown in Fig 1C, and forecast results of all cities in Guangdong in 2019 for the whole year in week 10 were shown in S8 Fig. The posterior fittings well capture the epidemic curves. However, it is noteworthy that although the prediction model incorporating population movement data yields good posterior fitting, there is no definite causal relationship can be given between the spatial transmission of dengue and population migration.

In the next section of the research, we investigated whether key parameters could be accurately inferred from simulated synthetic epidemics. In Fig 3A and 3B, two such outbreaks were generated by the metapopulation networks with the same initial conditions and parameters except for the random movement index (A, $\theta = 0.05$ and B, $\theta = 0.25$). Afterward, we assimilated the simulated observations using the proposed system and obtained parameter inference results, and the posterior fitting generated by the system for assimilating simulated outbreaks were shown in S9 and S10 Figs. Fig 2 shows the average of 300 ensembles parameter inferences. The results show that these calibrated parameters have stabilized significantly around the true values after assimilating several weeks of observations. Only 6 cities were presented in the picture for clarity, and the complete results were presented in the Appendix (S11 Fig). At the same time, we calculated the relative error between the inferred and true values of the predicted target. These results suggest that the accuracy of the inference is not particularly sensitive to the initial conditions of the model. To further validate the stability of the model parameter inferences, we compared the prediction performance of the isolated SIR-EAKF model and the metapopulation SIR networks-EAKF system for 10 synthetic outbreaks, and the results were presented in the S12 Fig. Our analysis results demonstrated the stability of the metapopulation model in terms of parameter inference. In addition, the analysis results of simulated outbreaks showed that the combined model significantly outperformed the isolated model with respect to the prediction accuracy.

However, dengue cases tend to occur sporadically based on historical and theoretical experience. For the simulated dengue outbreaks, the metapopulation model simultaneously integrates dengue cases from multiple subpopulations and rounds the simulated values to integers. The rounding process may miss some scattered cases, resulting in a slight underestimation. Therefore, random numbers sampled from a Poisson distribution need to be added to the dengue outbreak simulations to obtain observations that are more consistent with real dengue outbreaks.

To assess the prediction accuracy of the model, we retrospectively predicted dengue cases in cities of Guangdong province in 2018–2020. The time series of dengue fever from 2015 to 2020 were shown in Fig 4A. Importantly, because the actual peak week and epidemic magnitude are unknown in real time, we assessed the forecast accuracy relative to the predicted peak week or epidemic magnitude in the prediction. We first defined the forecast accuracy for the

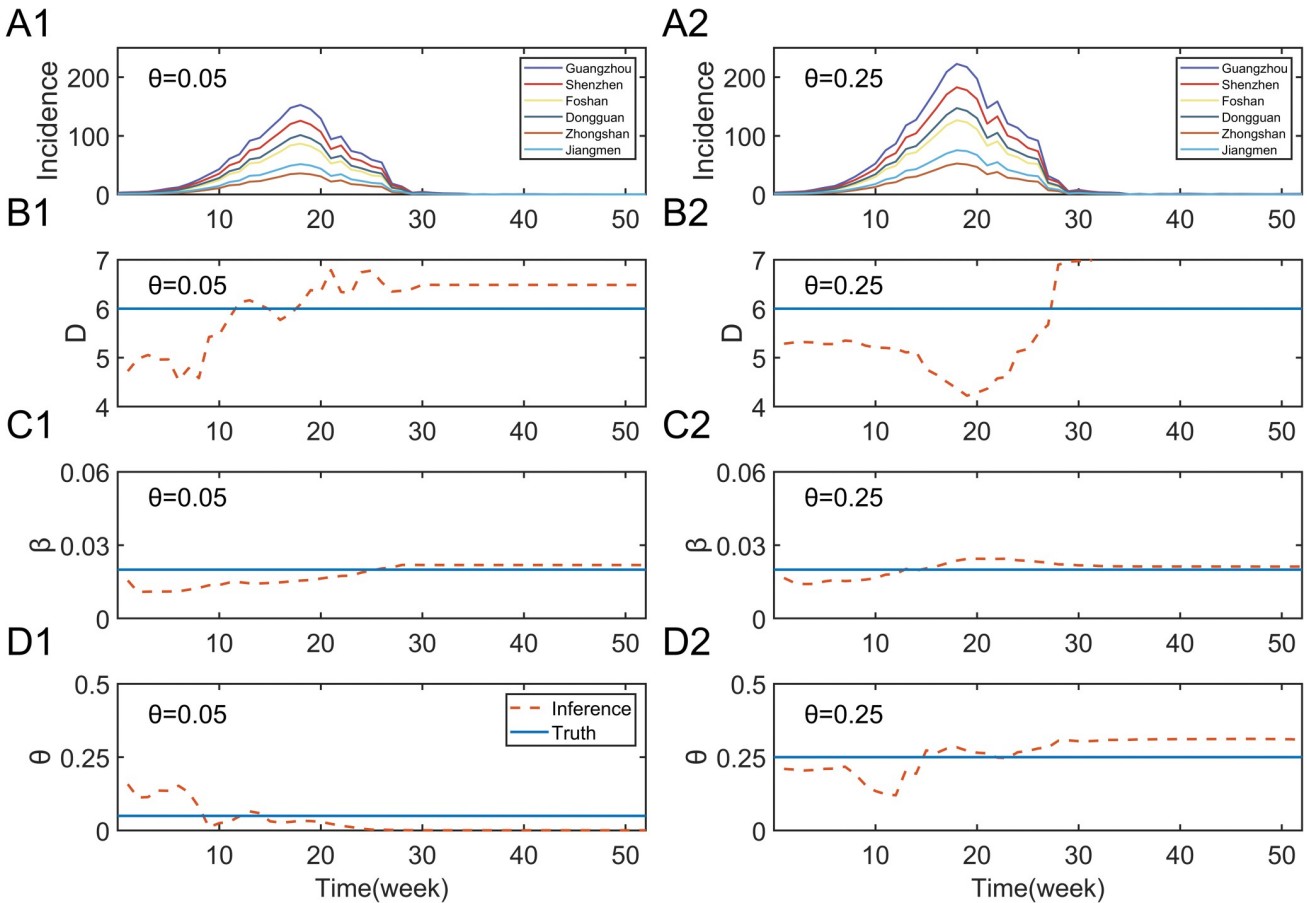

**Fig 3. Inference of parameters in the metapopulation model.** The metapopulation model generates the number of new infections per week in six cities (Guangzhou, Shenzhen, Foshan, Dongguan, Zhongshan, and Jiangmen). Except (A1) $\theta = 0.05$ and (A2) $\theta = 0.25$, both simulations were obtained under the same initial conditions. Different colors distinguish the epidemic curves of new cases in different cities. (B1: $\theta = 0.05$; B2: $\theta = 0.25$) Inference of the parameter D (infection duration) by the metapopulation network-EAKF fitting. The real blue line represents the real parameters used in the simulated epidemics, and the red dotted line represents the posterior mean in the data assimilation process. (C1: $\theta = 0.05$; C2: $\theta = 0.25$) The parameter $\beta$ (contact rate) in the metapopulation model is inferred. (D1: $\theta = 0.05$; D2: $\theta = 0.25$) Inference of the parameter $\theta$ (random movement ratio) in the metapopulation model.

peak week as the fraction of the average ensemble forecast within ±1 week from observations; the forecast accuracy for the peak intensity as the fraction of the average ensemble forecast within ±25% from observations; the forecast accuracy for the total incidence as the fraction of the average ensemble forecast within ±25% from observations. The average prediction accuracy of peak week, peak intensity, and total incidence of dengue fever in Guangdong province for the 2019–2020 dengue season were shown in Fig 4B, respectively. In addition, the predictions for the 2018–2019 dengue season were shown in S13 and S14 Figs. The horizontal axis of the graph represented the number of weeks ahead of the forecast. A positive forecast lead indicated that the starting week or peak week was expected to occur in the future, while a negative value meant that the starting week or peak week was expected to have passed. The accuracy of the combined model based on metapopulation networks and EAKF algorithm for predicting the peak week ten weeks in advance was more satisfactory. We also found that the model proposed in our study had higher prediction accuracy of infection, although results varied greatly across prefecture-level city. Specifically, we also summarized and compared the prediction accuracy of the two models for peak week in each city. We found that the established model

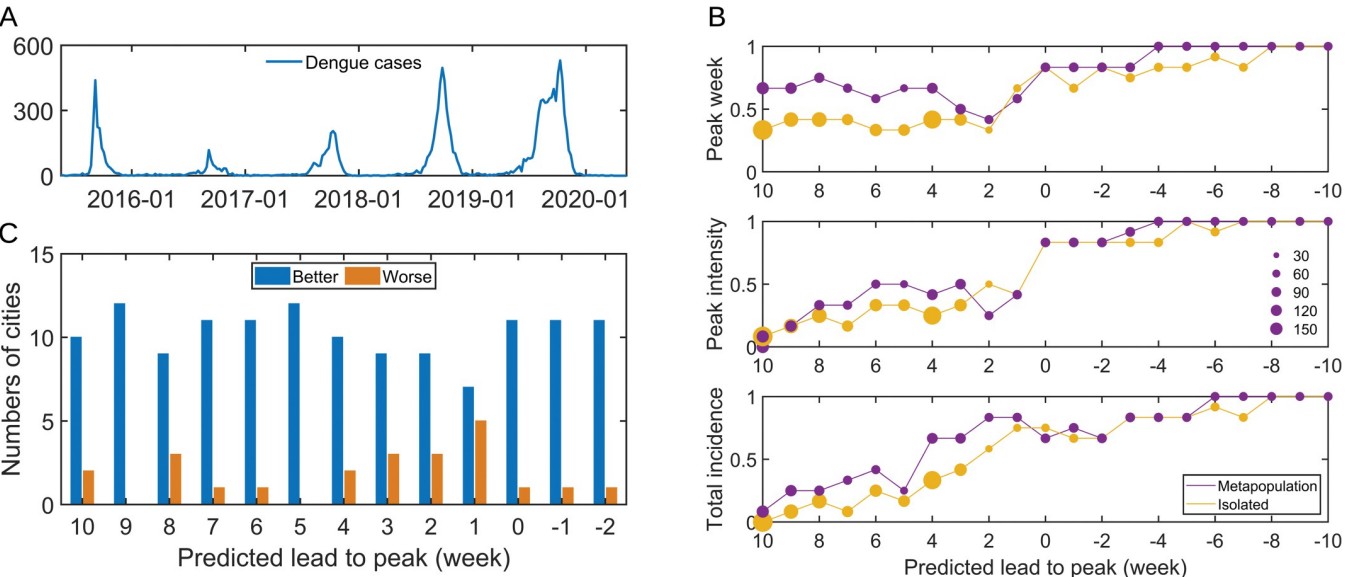

**Fig 4. Prediction of spatial transmission of dengue fever in Guangdong province, China.** (A) The epidemic situation of dengue fever in Guangdong province. (B) The average prediction accuracy of three evaluation metrics (peak week, peak intensity, and total incidence) for metapopulation and isolated predictions across 21 cities in 2019. (C) The performance of prediction of peak week using metapopulation model compared with the isolated model in individual cities. Blue (orange) bars indicate that the metapopulation forecast is better (worse) in terms of the number of cities.

outperformed the isolated model for most cities in the aspect of prediction accuracy. In particular, the proposed model demonstrated good prediction performance for most cities with leads of up to 10 weeks. Table 1 summarized the improvements in predictive accuracy for peak week, peak intensity, and total incidence and the statistical significance obtained from bootstrap analysis. The improvements were generally statistically significant indicating significantly improved prediction accuracy for peak week and total incidence. The prediction of peak intensity was relatively more challenging, but the prediction accuracy of the metapopulation model was still better than that of the isolated model, although the improvement in prediction accuracy at sporadic weeks was not statistically significant.

Our results show that the predicted number of dengue cases predicted by the model proposed in this study was closer to the true number of dengue cases and that the predictions of the ensemble forecast system were always correct when both models predicted simultaneously. From the perspective of model application, the SIR-EAKF system combined with population movement data can retrospectively infer the spatial spread of the virus. Further, surveillance data from different cities can be used by the model to simultaneously generate estimates of the outbreak magnitude, thus avoiding inaccurate local reporting data due to underreporting, which was beneficial for predicting both novel and currently existing viruses.

**Table 1. Accuracy of metapopulation predictions improvement compared with the isolated forecast of peak week, peak intensity, and total incidence.**

| Target | Predicted lead to peak | | | | |
|---|---|---|---|---|---|
|  | 6 weeks | 5 weeks | 4 weeks | 3 weeks | 2 weeks |
| Peak week | 11** | 12** | 10** | 9** | 9** |
| Peak intensity | 7 | 8* | 9** | 7 | 8* |
| Total incidence | 6 | 8* | 8* | 10** | 9** |

Asterisks indicate the statistical significance of bootstrap analysis: $0.01 <^* P < 0.05$, $^{**} P < 10^{-5}$.

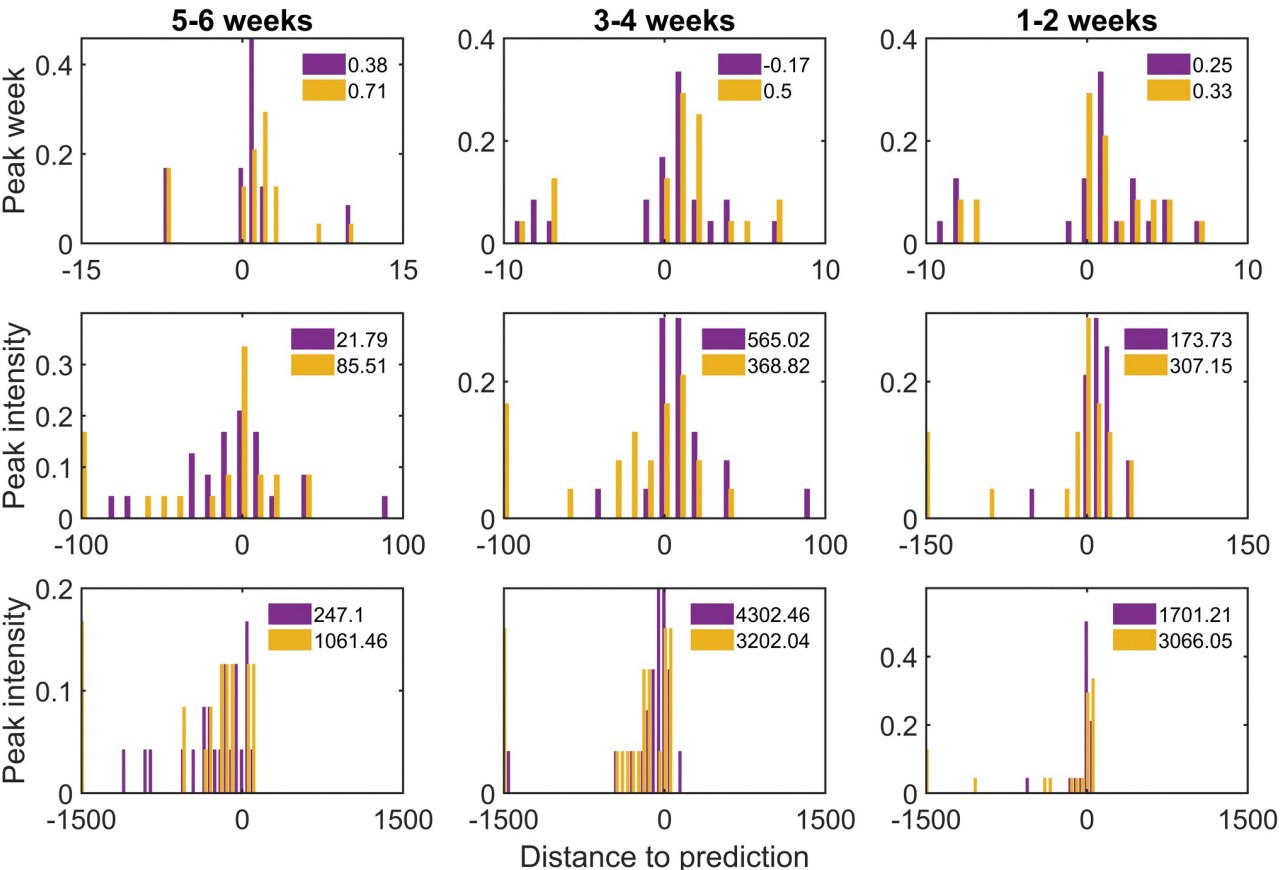

**Fig 5. Distribution of the distance from the observed target to the predicted value.** Blue columns represent metapopulation forecasts, and orange columns represent isolated forecasts. The mean deviations of each distribution are displayed in legends of each subgraph and distinguished by color. (top) For peak forecasts with predicted leads of 5 to 6, 3 to 4, and 1 to 2 wk, the distributions of the peak week relative to the predicted value (x-axis is the observed peak week minus the predicted peak week) in 21 cities in 2019 are shown, both for metapopulation and isolated prediction. (middle and bottom) The same analysis for (middle) peak intensity and (bottom) total incidence, grouped by the predicted lead to peak week. The x-axis for middle (bottom) is the value of observed peak intensity (total incidence) minus predicted peak intensity (total incidence).

The above comparison of the prediction accuracy of the results was performed based on observations. However, these observations inherently contain a degree of randomness in the actual predictions. Therefore, researchers have viewed each observation as a single realization of a stochastic process [36]. To further validate the forecasting system, we evaluated the forecasting results by exploring the distribution of observations relative to the predicted values in each forecast. Specifically, when the observations were normally distributed around the predicted values with zero mean, the forecast provides a good estimate of the true value. Conversely, a bias in the prediction can be perceived when the mean error was not zero. Therefore, estimates of forecast inaccuracy were obtained by exploring the distribution of observations around the forecast. Fig 5 presents the distance distributions from the observed to the predicted values for the three indicators (peak week, peak intensity, and total incidence) for 12 prefecture-level cities in Guangdong province in 2019. We grouped the model-generated forecasts by forecast week from 6 weeks to 1 week (peak week, peak intensity, and total incidence) and then the distance distribution of the observed values relative to the predicted values under each category was plotted separately. The relative distances predicted by the two models for the predicted targets (peak weeks, peak intensity, and total incidence) are shown in the legends of Fig 5. Unlike the metapopulation networks, isolated forecasts were more sensitive to

observation noise. As for the metapopulation networks, the identification of early signals of the epidemic was extracted from neighboring cities due to the introduction of population movement data, which greatly reduced the influence of the model from zero-inflated data. Meanwhile, most of the metapopulation networks in the results had a smaller mean bias, suggesting that the proposed model demonstrated a better prediction performance than isolated model. However, we found that early sporadic observations may affect the model and may cause the isolated model prediction system to incorrectly predict peak weeks, as indicated by the bias in the peak week prediction (Fig 5, top). For peak intensity and total incidence, the distribution of isolated model predictions tended to be positive skewed, especially for predicting total incidence (Fig 5, bottom), implying that the isolated model failed to capture the outbreak trajectory in low-disease periods.

## Discussion

Here we have developed and retrospectively validated an outbreak prediction system that can accurately predict the epidemic size and temporal peak of dengue fever. Given dengue infection cases and population movement data, the metapopulation network-EAKF system iteratively calibrated state variables and model parameters to provide accurate prediction results and accurately corrected model parameters in real time. By assimilating surveillance data from multiple cities, the metapopulation networks improved prediction accuracy for peak week, peak intensity, and total incidence of dengue outbreaks. Our models projected that there were 12, 9, and 10 cities with improved predictions of the predicted targets compared to baseline predictions made at each city individually, and these cities accounted for 100%, 75%, and 83.3% of the total number of cities predicted, respectively.

In this study, the isolated model in Guangzhou city is consistent with most of the work done by the SIR-EAKF theory [26]. The metapopulation network combined with population migration information can improve the prediction accuracy compared with the isolated model. These results appear to be consistent with other studies showing that human mobility has expanded the spatial range of transmission, leading to changes in the amount of contact between susceptible populations and infected mosquitoes [15,38–41]. Since this study includes the spatial network in the meta-population model, it can well solve the problem of dengue transmission from one city to another, which has not been solved in previous studies [17,20]. At the same time, our results confirm the view of Jeffrey Shaman [16], who believes that larger areas may be better predicted if they are divided into smaller geographical units. These issues need to be explored more comprehensively in the future to determine the optimal spatial scale at which dengue should be predicted.

This study represents the first attempt to establish a combined system based on metapopulation networks and Kalman filter using population mobility data. The system achieved good predictions of the spatial spread of dengue fever and retrospectively validated the improvement of the model over the baseline model through multiple validation methods. Given that the spatial network-based approach of our study relies only on dengue case count data routinely collected in most endemic countries, we anticipate that similar models could be applied in many other countries globally. In addition, since surveillance data of dengue fever were derived from a passive detection system, it was difficult to monitor mild patients [42,43] or asymptomatic patients [22,44], which may lead to underreporting bias. The prediction system proposed in this study can capture the signal of the occurrence of the epidemic in the early sporadic case period of dengue fever by absorbing surveillance data from multiple cities and give satisfactory predictions in advance, which can minimize the impacts of underreporting bias.

Nevertheless, our study has several major limitations. First, although the model uses mobile phone data provided by China Unicom to estimate population mobility data in Guangdong province, there was a certain proportion of children among dengue patients [45]. Thus, the information of these children or the elderly may not be recorded by mobile phone data, and there may be some underrecording of data records. However, a random mobile population parameter was added to our model to consider the random occurrence of population movements, which can avoid this bias to some extent. Next, this study examined the spread of dengue fever on a large scale based on population movements between cities. However, the study could not determine the subtle differences between the actual travel distance of individuals, the time spent in different locations, and the frequency of specific types of habitats (e.g., residential, entertainment, and business) [40]. Third, the question of whether population mobility or climatic factors were the only causes of differences in onset dates among cities cannot be resolved yet. However, the predictive accuracy of our model suggests that it broadly captures the relationship between mobility and transmission, and we thus expect our broad conclusions. For example, the metapopulation model has higher prediction accuracy in part because it can obtain signals from the population mobility network, but the isolated model cannot do that. It was hoped that work will stimulate clear causal relationship studies among these factors in further research. Forth, the significant change in human mobility following the lockdown in late January 2020 was not captured in our model. However, the dengue season had essentially ended in Guangdong province from January to May 2020. This time period had limited dengue cases with or without COVID and should have a limited impact on the overall results.

The prediction system developed in this study incorporated data assimilation algorithms that differ from previous disease-specific transmission dynamics, so it may be applied to the prediction of other infectious diseases. In recent decades, many new infectious diseases have emerged rapidly and spread globally, including influenza, Zika virus, chikungunya fever, Ebola virus, avian influenza, and coronavirus disease. Our forecast system naturally extends to other disease, mobility datasets, models, and algorithms that capture more aspects of real-world transmission, and these represent interesting directions for future work. In an increasingly interconnected world, providing policy-relevant and real-time information on when and where dengue outbreaks will occur, and how to effectively intervene, monitor, and clinically respond can provide useful evidence-based information for policy development and control coordination. The forecasting system proposed in this study combined complex ensemble population models, efficient data assimilation techniques, mobile phone data, large-scale climate data [46], mosquito density surveillance data [47,48], and dengue cases surveillance information at the prefectural city level, which can be performed in real time to generate forecasts of future disease transmission and magnitude. Accurate forecast of the intensity and temporal peak of infectious disease outbreaks discriminated among cities or regions within a country would provide greater lead time for preferential focus of mitigation and response resources to areas with more urgent need. In conclusion, the network structure exploration is helpful for targeting risk actors and tailoring prevention and control strategies. This study demonstrates the potential and challenges of spatio-temporal models in improving the understanding of dengue spatial transmission, and provides valuable empirical evidence for government public health departments to guide the refinement of vector control strategies.

## Supporting information

**S1 Text Method. The association of climate, mosquito vectors, and population movements with the occurrence of dengue fever, analysis of the intercity population mobility patterns, temperature and mosquito vector-driven SIR model for dengue fever, data assimilation**

**methods, and references.**
(DOC)

**S1 Fig. Weekly dengue fever cases (cross-sign) in each city of Guangdong Province in 2018.**
(TIF)

**S2 Fig. Weekly dengue fever cases (cross-sign) in each city of Guangdong Province in 2019.**
(TIF)

**S3 Fig. Distribution of relative inference errors.** For θ = 0.05 (left) and θ = 0.25 (right), the relative error distribution of D, β, and θ in the 21st week was deduced by 300 independent estimations using the metapopulation model. Each estimate is the mean of the randomly selected 300 member sets.
(TIF)

**S4 Fig. Dengue curve of 21 cities in Guangdong province from 2018 to 2020.** The retrospective forecast season is marked by grey areas.
(TIF)

**S5 Fig. The distribution of movement in adjacent and non-adjacent cities.**
(TIF)

**S6 Fig. The association of climate, mosquito vectors and population movements with the occurrence of dengue fever.** (A) weekly observations of dengue fever, temperature, Mosquito Oviposition Index (MOI), and population movements for each prefecture-level city in Guangdong Province from 2018 to 2020. (B) Climate, population movement data and mosquito vector MOI index drive dengue incidence, and Cross mapping skills were calculated for each city in Guangdong.
(TIF)

**S7 Fig. Posterior predictions of dengue fever in Guangdong province in 2018.** Weekly dengue fever cases (cross symbols) for each prefecture-level city. The solid lines and shaded areas are the posterior mean and 95% credible intervals (CI), respectively, of the metapopulation network-EAKF fit.
(TIF)

**S8 Fig. Forecast of epidemic curves in 21 cities of Guangdong province in 2019.** The metapopulation model with 300 ensemble members is used. The yellow cross symbols indicate weekly observations. The solid red curves are average forecast trajectories, and the grey areas represent 95% prediction intervals.
(TIF)

**S9 Fig. Inference of parameters in the metapopulation model.** Simulations were obtained under the same initial conditions, and the random movement parameter is set as θ = 0.05. Posterior mean estimates of weekly observation of new cases in metapopulation simulations are displayed for 21 cities separately. The solid line and shadow area are the posterior mean and 95% confidence interval (CI) of the metapopulation network-EAKF fitting, respectively. The red cross symbols indicate the synthetic observations used for data assimilation.
(TIF)

**S10 Fig. Inference of parameters in the metapopulation model.** Simulations were obtained under the same initial conditions, and the random movement parameter is set as θ = 0.25. Posterior mean estimates of weekly observation of new cases in metapopulation simulations are displayed for 21 cities separately. The solid line and shadow area are the posterior mean and

95% confidence interval (CI) of the metapopulation network-EAKF fitting, respectively. The red cross symbols indicate the synthetic observations used for data assimilation.
(TIF)

**S11 Fig. Inference of parameters in the metapopulation model.** (A and B) The metapopulation model generates the number of new infections per week in 21 cities. Except (A) $\theta = 0.05$ and (B) $\theta = 0.25$, both simulations were obtained under the same initial conditions. Different colors distinguish the epidemic curves of new cases in different cities. (C $\theta = 0.05$; D $\theta = 0.25$) Inference of the parameter D (infection duration) by the metapopulation network-EAKF system. The real blue line represents the real parameters used in the simulated epidemics, and the red dotted line represents the posterior mean in the data assimilation process. (E $\theta = 0.05$; F $\theta = 0.25$) The parameter $\beta$ (contact rate) in the metapopulation system is inferred. (G $\theta = 0.05$; H $\theta = 0.25$) Inference of the parameter $\theta$ (random movement ratio) in the metapopulation model.
(TIF)

**S12 Fig. Average forecast accuracy of peak week, peak intensity, and total incidence for pooled population and isolated forecasts.** Based on population movement, ambient temperature, and Mosquito Oviposition Index data from 21 cities in Guangdong, 10 synthetic outbreaks were generated using an ensemble population model. For each synthetic outbreak, weekly forecasts were made using 300-member ensembles.
(TIF)

**S13 Fig. Average prediction accuracy for metapopulation and isolated predictions of peak week, peak intensity, and total incidence for the 2018 dengue fever season. y-axis scale indicates the proportion of accurate predictions in each group.**
(TIF)

**S14 Fig. Two forecast systems predicted the integrated epidemic curve of seven cities at week 10.** Two forecast systems predicted the integrated epidemic curve of seven cities in week 10. Research provides prediction methods using metapopulation model (A) and isolated model (B) prediction systems. Three hundred ensemble members are used in the forecast. 95% prediction intervals (PIs) are reported. The metapopulation forecast system well predicts the outbreak curve. Although the Isolated model captured seasonal targets such as peak week in a few locations, the overall epidemic curve is underestimated by isolated prediction.
(TIF)

**S1 Table. The number of cases predicted by metapopulation and isolated model before peak week.**
(DOCX)

**S1 Video. Daily population movements among prefecture-level cities in Guangdong in 2018.**
(MOV)

## Acknowledgments

We thank the staff members at the hospitals, local health departments, and county-, district- and prefecture-level Centers for Disease Control and Prevention for their great assistance in coordinating data collection. We are also thankful to the staff of the China Unicom Company for their tremendous help in mobile phone data collection.

## Author Contributions

**Conceptualization:** Qinghui Zeng.

**Data curation:** Qinghui Zeng.

**Formal analysis:** Qinghui Zeng, Haobo Ni.

**Funding acquisition:** Jianpeng Xiao, Pi Guo.

**Investigation:** Qinghui Zeng, Xiaolin Yu.

**Methodology:** Qinghui Zeng, Yuliang Chen, Sen Pei, Pi Guo.

**Project administration:** Pi Guo.

**Resources:** Hui Deng, Yingtao Zhang, Jianpeng Xiao, Pi Guo.

**Software:** Qinghui Zeng.

**Supervision:** Xiaolin Yu, Lina Xiao, Ting Xu, Haisheng Wu, Yuliang Chen, Hui Deng, Yingtao Zhang, Sen Pei, Jianpeng Xiao, Pi Guo.

**Validation:** Pi Guo.

**Visualization:** Qinghui Zeng.

**Writing – original draft:** Qinghui Zeng.

**Writing – review & editing:** Qinghui Zeng, Xiaolin Yu, Haobo Ni, Lina Xiao, Ting Xu, Haisheng Wu, Yuliang Chen, Hui Deng, Yingtao Zhang, Sen Pei, Jianpeng Xiao, Pi Guo.

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
