## [Decision Letter · Decision Letter 0]

26 Sep 2022

Dear Dr. Guo ,

Thank you very much for submitting your manuscript "Dengue transmission dynamics prediction by combining metapopulation networks and Kalman filter algorithm" for consideration at PLOS Neglected Tropical Diseases. As with all papers reviewed by the journal, your manuscript was reviewed by members of the editorial board and by several independent reviewers. In light of the reviews (below this email), we would like to invite the resubmission of a significantly-revised version that takes into account the reviewers' comments. 

The Authors are expected to address all the criticisms by all Reviewers. In particular, please describe the novel contribution of your model, and implications in terms of early warning or other interventions (Reviewers #1 and #2), further elaborate your model and meta-population structure (Reviewers #1 and #3), and clarify whether change in human mobility due to the COVID-19 pandemic has been considered or incorporated (Reviewers #2 and #3).

We cannot make any decision about publication until we have seen the revised manuscript and your response to the reviewers' comments. Your revised manuscript is also likely to be sent to reviewers for further evaluation.

Sincerely,

Eric HY Lau, Ph.D.

Academic Editor

Robert Reiner

Section Editor

The Authors are expected to address all the criticisms by all Reviewers. In particular, please describe the novel contribution of your model, and implications in terms of early warning or other interventions (Reviewers #1 and #2), further elaborate your model and meta-population structure (Reviewers #1 and #3), and clarify whether change in human mobility due to the COVID-19 pandemic has been considered or incorporated (Reviewers #2 and #3).

Reviewer's Responses to Questions

**Key Review Criteria Required for Acceptance?**

**Methods**

-Are the objectives of the study clearly articulated with a clear testable hypothesis stated?

-Is the study design appropriate to address the stated objectives?

-Is the population clearly described and appropriate for the hypothesis being tested?

-Is the sample size sufficient to ensure adequate power to address the hypothesis being tested?

-Were correct statistical analysis used to support conclusions?

-Are there concerns about ethical or regulatory requirements being met?

Reviewer #1: The manuscript entitled “Dengue transmission dynamics prediction by combining metapopulation networks and Kalman filter algorithm” applied mathematical model incorporated with EAKF approach to simulate dengue transmission in Guangdong. Overall, the models and results are sound; however, some detailed of modeling approaches should be clarified. 

Major issues:

1. The way to avoid zero-inflated issue is a little confusing. The author just shifted the initial time point to the 19th week rather than the first week of the year. However, the weekly data still contains many zeros. Could you explain more about why the changes can deal with the zero-inflated issue?

2. The author use real-time forecasting might not be appropriate term because you deal with weekly data. You need to define the meaning of “real time” in your study.

Reviewer #2: (No Response)

Reviewer #3: (No Response)

**Results**

-Does the analysis presented match the analysis plan?

-Are the results clearly and completely presented?

-Are the figures (Tables, Images) of sufficient quality for clarity?

Reviewer #1: Since the model can give more accurate forecasting results about “peak time”, “peak intensity” and “total number” of cases. I was wondering to know whether the model can be beneficial to the dengue early warning system?

Reviewer #2: (No Response)

Reviewer #3: (No Response)

**Conclusions**

-Are the conclusions supported by the data presented?

-Are the limitations of analysis clearly described?

-Do the authors discuss how these data can be helpful to advance our understanding of the topic under study?

-Is public health relevance addressed?

Reviewer #1: Regarding the 3rd limitation mentioned in the discussion, is it possible to develop sub-models to include either climate or mobility data only? If it doable, you can understand which parameters play more important role in your model.

Reviewer #2: (No Response)

Reviewer #3: (No Response)

**Editorial and Data Presentation Modifications?**

Reviewer #1: (No Response)

Reviewer #2: (No Response)

Reviewer #3: (No Response)

**Summary and General Comments**

Reviewer #1: 1. No page and line number make it difficult to pinpoint questions.

2. The vector’s name should write in this way: Aedes albopictus and Aedes aegypti

Reviewer #2: The authors present a novel method for more accurately tracking dengue virus disease dynamics by integrating a metapopulation network contact structure based on human mobility data inferred from cell phone use data, temperature, precipitation, and mosquito population dynamics. They utilize an ensemble-adjusted Kalman filter algorithm for data assimilation. Overall, the manuscript presents important findings on dengue analytic tools that will likely prove relevant in other geographic contexts and other disease contexts. However, the objectives of the study and its novel contribution are not well described. The research is not well-situated in the broader literature. More clarity on the relative contribution of this study and the implications for infectious disease dynamics modeling based on an understanding of the literature would strengthen this article's candidacy for publication. The first sentence of the discussion state's the main purpose of the research, but it is not clear how much of this purpose has been accomplished by previous studies, and this statement of purpose should come sooner in the manuscript. It is also unclear if data and code will be made available, making the reproducibility of the study difficult.

Abstract:

The first sentence says predicting the magnitude and timing of disease outbreaks is critical. But is that what this study really does? Does it predict where, when, and how bad novel outbreaks will be? Or does it rather improve accuracy of final epidemic size and when the epidemic will inflect, given outbreaks? Predicting the temporal peak of the epidemic is not the same as timing writ large.

The authors conclude the abstract with asserting that the paper provides a foundation for forecasting mosquito-borne infectious diseases. Is this claim supported? Will this model work with malaria or other mosquito-borne diseases that utilize different mosquito vectors with different habits/preferences? Aedes mosquitoes are importantly different than Anopheles.

Author summary:

The statement that human mobility increases the rapidity of geographic spread of dengue does not logically lead to the following statement that 'therefore, the establishment of models... is necessary for epidemic preparedness.' That would depend on how such models are used.

The authors state that 'this study developed and validated a forecasting system for predicting the spatial spread of dengue.' However, the model has not been used in predicting and forecasting, except retrospectively. This statement should include the timeframe of the data utilized.

Introduction 

Para 1

Please state the estimate of the economic loss, rather than use the subjective term 'enormous.'

Para 2

'Previous studies have proposed...'

Only one study is cited here. 

Para 2-3

It is not clear what, precisely, has been already done in the literature. Can you state more precisely what the literature has studied so far and what specifically is novel about this approach? It seems that previous studies have looked at time series data, but others have also used human mobility. And cell phone use is likewise not novel. Is the novelty in this study the use of the Kalman filter? Or are the metapopulation structures also novel? If, for example, the sole novelty of the study is the inclusion of the Kalman filter, then when testing its performance compared to previous methods, it would be interesting to see the same model structure without the use of the filter algorithm. How do the constituent parts of the model explain or influence disease dynamics and can we rank-order their influence? 

Methods

Why were small outbreaks excluded? Can the model not predict small outbreaks too? Including small outbreaks seems important if the ultimate goal is accurate forecasting for epidemic preparedness.

Is 'eggs / effective eggs' a list or an equation?

Multiple formula are mentioned but not delineated. Can you delineate the formula used for MOI and dengue virus reproduction associated with temp? We need to have the math in the paper in order to understand how the model is constructed.

If China Unicom data is used, and that represents only 19% of market share, how is mobility writ large scaled up? Do you assume all cell phone carriers have the same mobility? What about people without cell phones?

How have mobility restrictions due to COVID-19 affected mobility and migration in the 2020 data and modeling?

The specifics of the metapopulation are not given. Consider a map or figure to illustrate how the metapopulation is structured and how it interacts between constituent nodes. I would also like to see a theoretical figure showing the inclusion of different modules (e.g. the metapopulation, mosquito population, temp, etc.) to better understand visually how the model works.

Parameter Inference

What are the assumptions of the EAKF? And are these assumptions met in this case? Is there an assumption of independence, for example, and are these variables independent?

Figure 2

Consider utilizing different labeling, so we can see how the graphics are paired (e.g. A1, A2, B1, B2...). Further, given the similarities (theta) of all graphs on the left v. right, consider a table of graphics.

Figure 3C

Any mentions of significance should cite the statistical method used to determine. 

Rather than report 'better' or 'worse,' consider reporting the magnitude of the differentials and analyzing for significance.

More can be done in the discussion to explore the study's contribution to the literature.

Reviewer #3: (No Response)

PLOS authors have the option to publish the peer review history of their article (what does this mean?). If published, this will include your full peer review and any attached files.

Reviewer #1: No

Reviewer #2: No

Reviewer #3: No
---

## [Decision Letter · Decision Letter 1]

11 Apr 2023

Dear Dr. Guo ,

Thank you very much for submitting your manuscript "Dengue transmission dynamics prediction by combining metapopulation networks and Kalman filter algorithm" for consideration at PLOS Neglected Tropical Diseases. As with all papers reviewed by the journal, your manuscript was reviewed by members of the editorial board and by several independent reviewers. In light of the reviews (below this email), we would like to invite the resubmission of a significantly-revised version that takes into account the reviewers' comments. 

The Authors are expected to address all the criticisms by all Reviewers. In particular, please address all comments from Reviewer #3 from the previous round, consider the impact of significant mobility change in early 2020 and clarify the metapopulation model (Reviewer #3). In additional to these comments, please address:

1. EAKF may handle zero-inflated data, but the significant change in human mobility after the lockdown in late January 2020 was not captured in the model (e.g. suspension of transport across or within county/provinces, cancellation of cultural/sport events). This could be a study limitation. However, this period is expected to have limited dengue with or without COVID and should have a limited impact on the overall results.

We cannot make any decision about publication until we have seen the revised manuscript and your response to the reviewers' comments. Your revised manuscript is also likely to be sent to reviewers for further evaluation.

Sincerely,

Eric HY Lau, Ph.D.

Academic Editor

Robert Reiner

Section Editor

The Authors are expected to address all the criticisms by all Reviewers. In particular, please address all comments from Reviewer #3 from the previous round, consider the impact of significant mobility change in early 2020 and clarify the metapopulation model (Reviewer #3). In additional to these comments, please address:

1. EAKF may handle zero-inflated data, but the significant change in human mobility after the lockdown in late January 2020 was not captured in the model (e.g. suspension of transport across or within county/provinces, cancellation of cultural/sport events). This could be a study limitation. However, this period is expected to have limited dengue with or without COVID and should have a limited impact on the overall results.

Reviewer's Responses to Questions

**Key Review Criteria Required for Acceptance?**

**Methods**

-Are the objectives of the study clearly articulated with a clear testable hypothesis stated?

-Is the study design appropriate to address the stated objectives?

-Is the population clearly described and appropriate for the hypothesis being tested?

-Is the sample size sufficient to ensure adequate power to address the hypothesis being tested?

-Were correct statistical analysis used to support conclusions?

-Are there concerns about ethical or regulatory requirements being met?

Reviewer #2: (No Response)

Reviewer #3: (No Response)

**Results**

-Does the analysis presented match the analysis plan?

-Are the results clearly and completely presented?

-Are the figures (Tables, Images) of sufficient quality for clarity?

Reviewer #2: (No Response)

Reviewer #3: (No Response)

**Conclusions**

-Are the conclusions supported by the data presented?

-Are the limitations of analysis clearly described?

-Do the authors discuss how these data can be helpful to advance our understanding of the topic under study?

-Is public health relevance addressed?

Reviewer #2: (No Response)

Reviewer #3: (No Response)

**Editorial and Data Presentation Modifications?**

Reviewer #2: (No Response)

Reviewer #3: (No Response)

**Summary and General Comments**

Reviewer #2: (No Response)

Reviewer #3: According to the reviewers' comments, the authors have made major modifications in the revised manuscript. However, there are three reviewers in the light of “Response To Editor”. There are no response to reviewer #3. There are one problem that the author have explained irrationally, and there is some problems that the authors have not revised. It is suggested that major revision should be made before publication.

Additional comments:

1. Both reviewers #2 and #3 comment how travel restrictions due to COVID-19 have affected the pattern of population movement in the modeling. The response of authors is that “The retrospective prediction of the season included in this study ended in February 2020. At that time, the outbreak of COVID-19 has only occurred in Wuhan city of Hubei province, and the outbreak has not yet spread to Guangdong (the study area of our study). ” This response is unreasonable. Firstly, in lines 195 and 196, the dengue season is defined as the 19th week of the former year to the 20th week of the succeeding year. So the season included in this study should ended in May 2020 rather than February 2020. Secondly, Guangdong reported its first confirmed case of COVID-19 on January 20, and by February 29, the total number of confirmed cases had reached 1,349. So the opinion that “the outbreak has not yet spread to Guangdong” is wrong. Thirdly, due to the COVID-19 outbreak in China in 2020, all Chinese provinces issued travel restrictions in late January, including Guangdong Province. The authors are advised to reconsider this comment.

2. In subsection Metapopulation network, the metapopulation network model is problematic. In Equation (2), the authors ignored the infection caused by mosquitoes biting travelers from other subpopulations I_n^k.

3. In lines 302 and 303, “U was the dissemination rate constant over an outbreak”. This definition is a little confusing. Maybe it is more reasonable to define U as vertical infection rate.

4. In Equation (2), what are the values of \\mu_b(t) and \\tau(t)? Please explain them in detail.

5. In lines 328 and 329, what do S_{nM}^k and I_{nM}^k mean? In line 365, what does N_{nM} mean? Do they mean that mosquito could move between cities? If so, it contradicts the assumption on line 296. Please explain them in detail.

PLOS authors have the option to publish the peer review history of their article (what does this mean?). If published, this will include your full peer review and any attached files.

Reviewer #2: No

Reviewer #3: No
---

## [Decision Letter · Decision Letter 2]

24 May 2023

Dear Dr. Guo ,

We are pleased to inform you that your manuscript 'Dengue transmission dynamics prediction by combining metapopulation networks and Kalman filter algorithm' has been provisionally accepted for publication in PLOS Neglected Tropical Diseases.

Best regards,

Eric HY Lau, Ph.D.

Academic Editor

Robert Reiner

Section Editor

Well done and congratulations on the excellent work!

Reviewer's Responses to Questions

**Key Review Criteria Required for Acceptance?**

**Methods**

-Are the objectives of the study clearly articulated with a clear testable hypothesis stated?

-Is the study design appropriate to address the stated objectives?

-Is the population clearly described and appropriate for the hypothesis being tested?

-Is the sample size sufficient to ensure adequate power to address the hypothesis being tested?

-Were correct statistical analysis used to support conclusions?

-Are there concerns about ethical or regulatory requirements being met?

Reviewer #3: (No Response)

**Results**

-Does the analysis presented match the analysis plan?

-Are the results clearly and completely presented?

-Are the figures (Tables, Images) of sufficient quality for clarity?

Reviewer #3: (No Response)

**Conclusions**

-Are the conclusions supported by the data presented?

-Are the limitations of analysis clearly described?

-Do the authors discuss how these data can be helpful to advance our understanding of the topic under study?

-Is public health relevance addressed?

Reviewer #3: (No Response)

**Editorial and Data Presentation Modifications?**

Reviewer #3: The authors have made sufficient modifications according to the modification comments, and I suggest this manuscript can be accepted without further modification.

**Summary and General Comments**

Reviewer #3: (No Response)

PLOS authors have the option to publish the peer review history of their article (what does this mean?). If published, this will include your full peer review and any attached files.

Reviewer #3: No

---

## [Editor Report · Acceptance letter]

2 Jun 2023

Dear Dr. Guo ,

We are delighted to inform you that your manuscript, "Dengue transmission dynamics prediction by combining metapopulation networks and Kalman filter algorithm," has been formally accepted for publication in PLOS Neglected Tropical Diseases.

Best regards,

Shaden Kamhawi

co-Editor-in-Chief

Paul Brindley

co-Editor-in-Chief
